# Advancements in Reference Gene Selection for Fruit Trees: A Comprehensive Review

**DOI:** 10.3390/ijms25021142

**Published:** 2024-01-17

**Authors:** Shujun Peng, Irfan Ali Sabir, Xinglong Hu, Jiayi Chen, Yonghua Qin

**Affiliations:** 1Guangdong Provincial Key Laboratory of Postharvest Science of Fruits and Vegetables, College of Horticulture, South China Agricultural University, Guangzhou 510642, China; 836234065@stu.scau.edu.cn (S.P.); 15238328269@163.com (X.H.); chenjiayi98@stu.scau.edu.cn (J.C.); 2Key Laboratory of Biology and Genetic Improvement of Horticultural Crops (South China), Ministry of Agriculture and Rural Affairs, College of Horticulture, South China Agricultural University, Guangzhou 510642, China; irfanalisabir@sjtu.edu.cn

**Keywords:** fruit trees, reference genes, qRT-PCR

## Abstract

Real-time quantitative polymerase chain reaction (qRT-PCR) has been widely used in gene expression analyses due to its advantages of sensitivity, accuracy and high throughput. The stability of internal reference genes has progressively emerged as a major factor affecting the precision of qRT-PCR results. However, the stability of the expression of the reference genes needs to be determined further in different cells or organs, physiological and experimental conditions. Methods for evaluating these candidate internal reference genes have also evolved from simple single software evaluation to more reliable and accurate internal reference gene evaluation by combining different software tools in a comprehensive analysis. This study intends to provide a definitive reference for upcoming research that will be conducted on fruit trees. The primary focus of this review is to summarize the research progress in recent years regarding the selection and stability analysis of candidate reference genes for different fruit trees.

## 1. Introduction

Fruit trees, much like other edible plants, hold significant commercial value in the agricultural industry. Beyond their economic importance, certain fruit trees also offer medicinal and health benefits. These remarkable trees contribute to human well-being by providing essential components for survival, encompassing not only their fruits but also their roots, stems, leaves, flowers, and seeds [1]. Moreover, it is worth noting that the majority of fruit trees are characterized as woody plants. Their roots serve a crucial function in preserving water and soil, playing a pivotal role in mitigating soil erosion and preventing desertification. In the current landscape, China has ascended to the position of the largest producer and consumer of fruits. Among these, the top five fruit trees, in terms of both fruit output and planting area, are citrus, apple, pear, peach, and grape. In recent years, the field of molecular biology has made tremendous strides, culminating in the sequencing of genomes for numerous fruit tree species [2]. Upon obtaining large-scale whole genome sequences, a paramount area of research comes into focus: the elucidation of gene function. It is imperative to recognize that the regulation of gene expression stands as the linchpin governing the structure and function of cells. This critical process not only underpins cell differentiation and morphogenesis but also endows organisms with the remarkable versatility and adaptability that define life itself.

Genes behave differently in cells at different times, and in different places under different circumstances. Current methods of analyzing gene expression levels mainly include northern blot, gene microarray, and real-time quantitative polymerase chain reaction (qRT-PCR) [3]. qRT-PCR has been extensively used in the field of gene expression studies because of its several benefits, including its speed, easy use, high sensitivity and specificity, capacity for batch detection, and diversity of application [4,5,6,7,8]. When quantifying mRNA transcription levels in samples, discrepancies among samples can arise from variations in RNA yield and reverse transcription efficiency. To mitigate these inconsistencies and ensure accurate measurements, it is essential to incorporate housekeeping or reference genes into the analysis. These reference genes serve as internal controls, allowing for the normalization of errors stemming from differences in RNA quantity, quality, and reverse transcription efficiency across samples, as illustrated in Figure 1. This normalization process is crucial for effectively reducing errors and revealing meaningful variations in the specific expression of target genes [9,10].

Housekeeping genes are a group of genes that are essential for basic cellular functions and are expressed in most cells of an organism [11,12]. These genes are often referred to as “constitutive genes”, as they are expressed continuously and not subject to regulation by external stimuli or developmental signals. Housekeeping genes are responsible for maintaining the basic functions of cells, such as metabolism, energy production, and protein synthesis. Examples of housekeeping genes include structural proteins like actin (ACT) and tubulin (Tub), enzymes involved in cellular respiration like glyceraldehyde-3-phosphate-dehydrogenase (GAPDH), and ribosomal proteins involved in protein synthesis [13,14]. Housekeeping genes play irreplaceable roles in encoding histone genes, ribosome protein genes, various enzymes in biological metabolic pathways, and mitochondrial protein genes.

In the late 1970s, a transformative breakthrough in molecular biology brought to light two pivotal classes of RNA: transfer RNA (tRNA) and ribosomal RNA (rRNA). These RNA varieties were aptly termed steward genes due to their crucial roles in cellular maintenance [15]. For example, *18s rRNA* is used as the reference gene in many studies [16,17]. tRNA is also considered a housekeeping gene due to its highly structured nature, usually consisting of about 76 nucleotides arranged in a clover secondary structure containing three stem loops [18]. The products encoded by these genes assume a paramount role in upholding both the structural and metabolic functions within cells. Notably, these genes exhibit constitutive expression, signifying that they are consistently active throughout every stage of an organism’s growth, across diverse cell types, and within nearly all tissues and organs. What is more, the expression levels of these genes remain largely unaffected by external environmental conditions, underscoring their indispensable roles in the core cellular processes that govern life.

So far, scientists have not found a single gene that works universally for qRT-PCR studies. The goal is to have a gene that can be used across different samples to make the results more consistent, but it has been challenging to find one. In other words, the absence of such a gene continues to pose certain limitations and inadequacies [19,20]. Therefore, in experiments involving gene expression analyses, it is crucial for researchers to carefully choose internal reference genes that are appropriate for the specific experimental conditions, cell types, and tissues being studied. Reference genes, also known as housekeeping genes, serve as a baseline for normalizing gene expression data, helping to account for variability in RNA quantity and quality across samples.

This review offers a comprehensive exploration of three critical facets concerning the selection and assessment of endogenous reference genes in fruit trees. It places emphasis on the pivotal role of precise reference gene selection, elucidates methodologies for evaluating the stability of these endogenous reference genes, and delves into recent advancements in the realm of research dedicated to endogenous reference genes in fruit trees. The ultimate objective of this review is to provide an invaluable resource for researchers within the field of molecular biology, particularly those focused on fruit trees.

## 2. Selection of Endogenous Reference Genes

qRT-PCR serves as a prominent technique for assessing gene expression levels, offering the ability to analyze gene expression in various plant tissue parts and developmental stages. However, to ensure the accuracy and reliability of qRT-PCR results, it is imperative to consider factors such as the quality of extracted RNA, concentration of complementary DNA (cDNA), and efficiency of PCR amplification. To address these concerns, the inclusion of a suitably stable internal reference gene becomes crucial for normalization, thereby enhancing the precision of the test outcomes. The ideal internal reference gene should exhibit consistent expression across diverse tissues within the organism and under varying physiological conditions, remaining unaffected by experimental variables [21].

The domestic caretaker gene, typically recognized as the internal reference gene, holds significant importance in preserving biological structure and facilitating metabolic processes. It is ubiquitously present in diverse tissues and cells, making it a suitable candidate for internal reference gene selection. The utilization of the domestic caretaker gene as an internal reference gene contributes to the accurate normalization of gene expression data across various experimental conditions and tissues, enhancing the reliability and robustness of qRT-PCR analyses [22].

### 2.1. Ideal Internal Reference Gene

An ideal reference gene must satisfy four fundamental conditions: (1) Stable expression across different tissues, organs, and cells within the same biological material or organism. This ensures consistent expression levels regardless of the specific tissue type, organ, or cell line, even during different growth and development stages. It enables reliable normalization across various sample sources [23]. (2) Insensitivity to biotic or abiotic stresses and other environmental factors, such as temperature, light, water, etc. The ideal reference gene should exhibit minimal expression variation in response to stresses and environmental conditions. This stability guarantees accurate normalization unaffected by external factors [24]. (3) Absence of pseudogenes and avoidance of genomic DNA amplification. It is essential to select a reference gene that does not possess pseudogenes, which could lead to inaccurate measurements. Additionally, precautions should be taken to avoid amplification of genomic DNA, ensuring specificity for the transcript of interest. (4) Expression levels similar to those of target genes, without significant differences. The expression level of the reference gene should closely resemble that of the target genes under investigation. It is important to establish that any differences between the reference gene and the target genes are not statistically significant, ensuring appropriate normalization [25]. By adhering to these conditions, the chosen reference gene will fulfill the requirements of stability, insensitivity to environmental factors, absence of pseudogenes, and similarity in expression levels, thereby serving as a reliable internal control for accurate gene expression analysis.

Indeed, the quest for a single gene that fulfills all of the aforementioned characteristics for an ideal reference gene remains challenging. Extensive evidence suggests that reference genes may exhibit differential expression stability under diverse experimental conditions. Consequently, it becomes imperative to thoroughly screen and select appropriate reference genes tailored to specific experimental contexts or employ combinations of multiple reference genes simultaneously. Neglecting to address the issue of inadequate expression stability among reference genes can significantly compromise the accuracy and reliability of experimental findings. By exercising careful consideration of experimental conditions and employing diligent reference gene screening methodologies, researchers can augment the robustness of their analyses, thereby ensuring heightened precision and trustworthiness of the obtained results.

### 2.2. Common Internal Genes

The housekeeping genes, commonly referred to as housesitter genes, occupy a pivotal position in molecular biology research. These genes perform essential functions that sustain minimal cellular activities, ensuring the overall viability of organisms. They exhibit ubiquitous expression across diverse cell types, underscoring their indispensable roles in fundamental biological processes. In investigations focusing on gene expression analyses in various plant species such as *Arabidopsis thaliana*, *Oryza sativa*, and *Zea mays*, the genes *actin* (*ACT*), *α-tubulin* (*α-Tua*), *β-tubulin* (*β-Tub*), *glyceraldehyde 3-phosphate dehydrogenase* (*GAPDH*), and *elongation factor-1α* (*EF-1α*), among others, have gained prominence as frequently employed internal reference genes [26,27,28,29]. Their consistent usage as reference points facilitates accurate normalization and reliable quantification of gene expression levels in a wide range of plant-based research endeavors (Table 1). However, related studies have shown that the expressions of these reference genes are not stable in plants with different cell types, tissue sites, and physiological states. Moreover, qRT-PCR requires high stability of reference gene expression, so the commonly used reference genes cannot meet its requirements.

## 3. Methods for Analyses of Internal Reference Gene Stability

Evaluating the stability of reference gene expression requires a meticulous assessment customized to the particular species and experimental conditions at hand. This process ultimately empowers researchers to identify genes exhibiting consistent expression patterns, making them ideal candidates as reference genes. To achieve this, it is essential to apply well-established criteria to scrutinize and compare their stability within the context of qRT-PCR analyses. Renowned software tools, such as BestKeeper, geNorm, and NormFinder, are widely employed for comprehensive analyses of reference gene expression variation and stability. These sophisticated computational solutions provide robust algorithms that facilitate meticulous assessments, empowering researchers to confidently identify reliable reference genes for precise normalization in gene expression investigations.

### 3.1. BestKeeper

BestKeeper (http://www.Gene-quantification.de/best-keeper.html) (accessed on 1 July 2023) is a software for internal reference gene and target gene expression analysis [30]. The software screening of reference genes operates on a systematic principle. It begins by conducting a paired correlation analysis of the sample, thereby assessing the interrelationships between different genes. This analysis yields important statistical parameters, including the standard deviation (SD), coefficient of variation (CV), and paired correlation coefficient (Poisson correlation coefficient, r). By meticulously comparing the magnitudes of these parameters, the software discerns internal reference genes that exhibit robust stability. This rigorous evaluation process enables the identification of optimal reference genes for precise normalization in gene expression investigations, thereby ensuring the accuracy and reliability of experimental results [31].

According to Wang et al. [31], the evaluation principle involves considering reference genes with an SD of less than 1 as stably expressed genes. Moreover, reference genes with a larger r, smaller SD, and smaller CV are deemed more stable, while those with lower values indicate reduced stability. BestKeeper, a software tool, not only facilitates the analyses of expression stability for internal reference genes but also enables comparative analyses of target gene expression levels. This comprehensive functionality empowers researchers to assess and compare the stability of both reference and target genes, thereby enhancing the accuracy and reliability of gene expression analyses. Despite its utility, an inherent limitation of the mentioned software is its restricted capacity to compare the expression levels of a maximum of 10 reference genes and 10 target genes in 100 samples. This constraint poses a challenge when working with larger datasets or when more extensive comparisons are required.

### 3.2. geNorm

The geNorm software, developed by Vandesompele et al. in 2002 and available for download at https://genorm.cmgg.be/ (accessed on 1 July 2023), serves a crucial purpose in qRT-PCR by screening reference genes and determining the optimal number of reference genes [9]. Its principle operates as follows:

First, the software compares the expression level of an internal reference gene with other internal reference genes through pairwise and logarithmic conversion. This process calculates the mean standard deviation, known as the Average Expression Stability Value (M), which reflects the gene’s expression stability. The software screens out internal reference genes with good stability, following the criterion that smaller M values indicate better stability, while higher values imply poorer stability. The default trade-off value for M in the software is set at M = 1.5. If an internal reference gene’s value is below 1.5, it is considered suitable as a reference gene.

Simultaneously, the software sequences the expression stability of all candidate reference genes and determines the optimal number of required reference genes based on paired difference analyses of standardized factors. geNorm introduces the pairwise variation value of a new reference gene and utilizes the V_n_/V_n+1_ ratio to determine the optimal number of reference genes. The default V value is 0.15, though it can be slightly adjusted. If V_n_/V_n+1_ is < 0.15, it indicates that n genes are sufficient as optimal reference genes. However, if V_n_/V_n+1_ is > 0.15, it suggests that the combination of n genes is not very stable, and the introduction of the n + 1 gene will significantly enhance the stability of the reference gene combination, thus necessitating the inclusion of the n + 1 gene. This software has proven instrumental in enhancing the accuracy and reliability of gene expression studies, contributing to more robust findings in the field of molecular research [32]. Indeed, the geNorm software’s versatile procedure can be employed to screen any number of reference genes in various experiments. By systematically evaluating the expression stability of potential reference genes and identifying the most stable combinations, researchers can select two or more reference genes to correct their data effectively. This approach significantly enhances the accuracy and reliability of relative quantitative results, thus empowering researchers to obtain more robust and precise findings [33].

### 3.3. NormFinder

The NormFinder software (http://www.mdl.dk/publicationsnormfinder.htm) (accessed on 1 July 2023) serves as a valuable resource for researchers to seek stable internal reference genes [34]. By utilizing an analysis of variance approach, this software enables a comprehensive assessment of the expression stability among candidate internal reference genes, ranking them based on their stability values. The most suitable reference gene is then selected, guided by the criterion that the one with the lowest expression stability value is deemed optimal [35].

Moreover, NormFinder goes beyond mere stability assessment and can also compare expression differences among candidate reference genes while calculating variations between sample groups. However, it is worth noting that NormFinder has a limitation since it can only identify a single most suitable reference gene [36]. Nonetheless, despite this constraint, the software remains an indispensable tool for researchers, significantly enhancing the accuracy and reliability of gene expression studies by facilitating the selection of a highly stable reference gene, thereby contributing to more credible and precise results. Indeed, the various software tools mentioned above employ distinct statistical analysis methods to identify suitable internal reference genes. Due to their different algorithms, the optimal internal reference genes identified by each software tool may not necessarily coincide.

To address this, RefFinder software (https://blooge.cn/RefFinder/) (accessed on 1 July 2023) serves as an invaluable tool for researchers, enabling a comprehensive analysis to obtain a comprehensive ranking index [37]. This index serves as a collective measure, taking into account the outputs from multiple software tools. The smaller the index value, the more stable the internal reference gene is deemed to be [38].

By integrating the results from different algorithms and software tools, RefFinder facilitates a more robust and well-rounded assessment of internal reference genes [39]. This comprehensive ranking index is crucial for researchers in selecting the most reliable and suitable internal reference genes, thus enhancing the accuracy and credibility of their gene expression studies.

## 4. Research Progress on Internal Reference Genes in Fruit Trees

### 4.1. Selection of Internal Reference Genes in Vegetative Organs

The process of selecting internal reference genes within the same organ can exhibit considerable variation across different species. As research delves deeper into this area, even within the same species, the diversity of endogenous reference genes has been revealed (Table 2). This observation underscores the complexity and species-specific nature of internal reference gene selection, necessitating meticulous consideration and evaluation to ensure the reliability and accuracy of gene expression studies within each unique biological context.

In the realm of plant research, the root holds paramount significance, driving a continuous exploration of stable internal reference genes across various species. In extensive studies conducted on diverse citrus varieties, the genes *18s rRNA*, *ACTB*, *RPII*, *IF3*, *Rpl35*, and *IF5A* emerged as consistently stably expressed candidates, making them well-suited as reference genes in deciphering gene expression levels in citrus roots [40,41]. Similarly, within the confines of the ‘Guanxi Honey pomelo’, researchers identified the *β-Tub* gene as exhibiting stable expression in its roots, thus making it a prime contender for selection as the internal reference gene [42].

In the case of peach roots from the *Prunus* genus, Tong et al. [43] made significant strides in identifying the *TEF2*, *UBQ10*, and *RPII* genes, which not only displayed stable expression but also demonstrated a moderate level of expression, rendering them as highly suitable internal reference genes for their research. Additionally, for a comprehensive understanding of key enzymes involved in betalain biosynthesis in pitaya, Chen et al. [44] conducted meticulous screening to ascertain endogenous reference genes in pitaya. Ultimately, *ACT(1)* emerged as the most fitting endogenous reference gene for their study.

These findings emphasize the critical importance of thoughtfully selecting appropriate internal reference genes, tailored specifically to the distinct plant species and organs under investigation. By doing so, researchers can ensure precise and reliable gene expression analysis, further advancing plant research endeavors.

In kiwifruit research, numerous studies have focused on identifying stable internal reference genes for accurate gene expression analyses in roots. Zhang et al. [45], and Zhou et al. [46] independently discovered distinct combinations of reference genes, such as *Tub* and *ACTB* combination and *ACT*, *GAPDH*, and *UBQ* combination, which exhibited consistent expression stability in kiwifruit roots, making them reliable internal reference gene candidates. In a groundbreaking effort, Liu et al. [47] accomplished the first selection of internal reference genes for figs. After a comprehensive analysis, they identified *18S rRNA* as the most suitable internal reference gene for gene expression analysis in fig roots. In the case of jujube roots, Meng et al. [48] successfully identified *ZjH3* as a stable internal reference gene, providing valuable insights for gene expression studies in jujube plants. These collective findings underscore the crucial importance of selecting species-specific stable internal reference genes, facilitating accurate and reliable gene expression analysis in the respective roots under study. Efforts to establish robust internal reference gene panels for specific plant species will contribute significantly to the advancement of scientific research in plant biology. 

In the context of stems and branches, the selection of internal reference genes exhibits considerable variability across different plant species. For instance, in the durian honey of the genus *Artocarpus*, *α-Tub1* and *β-Tub2* were identified as suitable internal reference genes [49]. In citrus, multiple studies have yielded varying reference genes such as *ACTB*, *18S rRNA*, *RPII*, *IF3*, *Rpl35*, and *IF5A* [40,41], while in pomelo, the reference gene turned out to be *β-Tub*, as revealed by Wang et al. [42].

Many studies on reference gene selection were reported in kiwifruit, with selected genes including *TUB* and *ACTB* combination and *ACT*, *GAPDH*, and *UBQ* combination [45,46,50]. In fig stems, *18s rRNA* emerged as the preferred reference gene [47], In grape, *RRM1* and *EF-1α* in combination are appropriate as internal reference genes [51]. For grape branch and leaf development processes, Ren et al. [52] identified *GAPDH*, *UBQ-1*, and *EF-1α1* as suitable reference genes. In the case of starfruit stems, Li et al. [53] determined *α-Tub* and *β-Tub* as the most fitting internal reference genes. Meanwhile, in jujube stems, Meng et al. found *ZjH3* to be a suitable endogenous reference gene for gene expression analysis in stem tips and fruiting branches [48].

These diverse findings emphasize the necessity of selecting species-specific internal reference genes tailored to the specific plant organs under investigation, enabling robust and accurate gene expression analyses in scientific research. Leaves, being pivotal sites for photosynthesis in plants, represent commonly used materials for investigating gene expression. Consequently, research efforts are centered on the selection of appropriate internal reference genes in leaves. In the context of peach leaves, *TEF2*, *UBQ10*, and *RPII* were identified as suitable reference genes [43]. The extensive research on kiwifruit has provided valuable insights, with Zhang et al. [45], Zhang et al. [50], and Zhou et al. [46] and Ferradás et al. [54] independently identifying *Tub* and *ACTB* combination, *ACT*, *GAPDH*, *UBQ*, *18s rRNA*, and *ACT 2* as appropriate internal reference genes in kiwifruit leaves. For fig leaves, the traditional reference gene *18s rRNA* was chosen as the appropriate internal reference gene [47], while in starfruit leaves, Li et al. [53] found *α-Tub* and *β-Tub* to be suitable reference genes.

The selection of internal reference genes in the leaves of jackfruit, durian, and jackfruit of different species varied. Guo et al. [49] and Wang et al. [55] identified *α-Tub*, *β-Tub*, *UBQ*, and *GAPDH* as suitable reference genes. In citrus leaves, *ACTB*, *RPII*, *18s rRNA*, *ACTB*, *FBOX*, *GAPC2*, *SAND*, *UPL7*, *IF3*, *Rpl35*, and *IF5A* were found to be appropriate reference genes [40,41,56]. However, a different internal reference gene, *β-Tub*, was identified in pomelo [42]. In pear leaves, it was concluded that *WDP* served as the most suitable internal reference gene [57], while *CYP2* and *RPII* were found to be suitable reference genes in leaves of litchi [58], respectively. *UQB* was also deemed suitable for the leaves of litchi [58] and apple [59]. For grape leaves, *GAPDH*, *UBQ-1*, *EF-1α1*, *RRM1*, and *EF-1α* combinations were found to be appropriate internal reference genes [51,52].

In jujube leaves, *ZjH3* was identified as the optimal internal reference gene [48]. During leaf development, the selection of internal reference genes changes. For instance, in apples, Pâmela Perin et al. [60] selected *MDH*, *SAND*, *THFS*, *TMp1*, and *WD40* as internal reference genes. In the development of grape leaves, Ren et al. [52] confirmed *18s rRNA*, *GAPDH*, *ACT*, *UBQ-1*, and *EF-1α1* as suitable reference genes for subsequent gene expression analyses.

The selection of internal reference genes for buds varies depending on the species. For leaf buds and flower buds of starfruit, Li et al. [53] identified *α-Tub* and *β-Tub* as suitable internal reference genes, while in jujube buds, Meng et al. [48] found *ZjH3* to remain an appropriate internal reference gene.

Furthermore, during the development of somatic embryos in longan, Lin et al. [61] selected *UBQ* and *Fe-SOD* combinations as appropriate internal reference genes. *UBQ* was also confirmed as a suitable internal reference gene in apple callus [59].

It is evident that even when testing gene expression in different organs of the same species, the chosen reference genes may not be uniform. Nevertheless, within plants of the same genus, the selected reference genes might exhibit partial similarity owing to their genetic closeness. Consequently, delving into the study of a particular species not only provides insights into that species but also establishes a foundation for researching other species. This approach proves beneficial in acquiring valuable information about additional species.

These findings underscore the importance of meticulously selecting appropriate species-specific internal reference genes when investigating gene expression in leaves and other plant tissues, ensuring precise and reliable results in scientific research.

**Table 2 ijms-25-01142-t002:** Selection of reference genes in vegetative organs of fruit trees.

Species	Genus	Vegetative Organs	Reference Genes	References
Apple	*Malus*	Leaves, callus	*UBQ*	[59]
		Leaf development process	*MDH*, *SAND*, *THFS*, *TMp1*, *WD40*	[60]
Pomelo	*Citrus*	Leaves, Stems, root	*β-Tub*	[42]
Citrus		Leaves	*ACTB*, *18S rRNA*, *RPII*	[40]
		Leaves, stems	*IF3*, *Rpl35*, *IF5A*	[41]
		Leaves	*FBOX*, *GAPC2*, *SAND*, *UPL7*	[56]
Pitaya	*Hylocereus*	Root, stems	*ACT(1)*	[44]
loquat	*Eriobotrya*	fruit setting	*GAPDH*, *UBCE*, *ACT*	[62]
		floral development	*GAPDH*, *EF1α*, *ACT*	
Durian honey	*Artocarpus*	Leaves, stems	*β-Tub2*, *α-Tub1*	[49]
Fig	*Ficus*	Leaves, stems, root	*18s rRNA*	[47]
Grape	*Vitis*	Branch and leaf development processes	*GAPDH*, *UBQ-1*, *EF-1α1*	[51]
		Leaves	*EF-1α*, *RRM1*	
		Tendril	*EF-1α* and *Actin* combination	[52]
		Leaves	*RRM1* and *EF-1α* combination	
Jackfruit	*Artocarpus*	Leaves	*UBQ*, *GAPDH*, *β-Tub*	[55]
Jujube	*Ziziphus*	Bud, fruiting branches, leaves, stem tips, root	*ZjH3*	[48]
Kiwifruit	*Actinidia*	Leaves, stems, root	*TUB* and *ACTB* combination	[50]
		Leaves	*18s rRNA*, *ACT 2*	[45]
		Leaves, stems, root	*GAPDH* and *UBQ* combination	[46]
Longan	*Dimocarpus*	Somatic embryo development process	*UBQ*, *Fe-SOD*	[61]
Lychee	*Litchi*	Leaves	*UBQ*, *RPII*	[58]
Peach	*Prunus*	Leaves, stems, root	*TEF2*, *UBQ10*, *RPII*	[43]
Pear	*Pyrus*	Leaf blade	*WDP*	[57]
Starfruit	*Averrhoa*	Leaf bud, leaves, stems	*α-Tub*, *β-Tub*	[53]

### 4.2. Selection of Internal Reference Genes in Reproductive Organs

The selection of reference genes in reproductive organs is wider than that in vegetative organs (Table 3). Fruit research has gained significant momentum, encompassing studies at both the physiological and molecular levels, with a particular focus on deciphering the underlying mechanisms. To delve into the intricate gene expression dynamics of fruits, it becomes imperative to measure and analyze gene activity accurately. In this pursuit, the identification of suitable internal reference genes plays a pivotal role. In the current scientific landscape, numerous studies have explored internal reference genes in fruits, at both the national and international levels. However, it is important to recognize that the choice of reference genes can differ across various fruit species and under diverse experimental conditions. As conscientious researchers, it is incumbent upon us to meticulously screen and select internal reference genes that align with the specific requirements of our experiments. This approach ensures the precision and reliability of subsequent gene expression analyses, contributing to robust and meaningful outcomes. In essence, the quest for ideal internal reference genes in fruit research represents a crucial step towards attaining scientific excellence and a deeper understanding of the intricate mechanisms governing fruit development and function.

In the screening of internal reference genes within the fruit pericarp, researchers have conducted thorough assessments and validations across various fruit species, including longan, apple, pear, banana, lotus mist, grape, and sweet cherry. Intriguingly, their findings have highlighted discrepancies in the choice of reference genes between different fruit pericarp types and their developmental stages, even within the same species. For example, Zhu et al. [62] found that the suitable internal reference genes for apple pericarp were *WD40*, *ACT* and *GAPDH*. Similarly, *GAPDH*, *Fe-SOD* and *Cu/Zn-SOD* were the suitable internal reference genes for longan pericarp while *Mn-SOD* and *EF-1α* need to be selected during the development of the pericarp [63]. Fan et al. [64] and Pâmela Perini et al. [60] found that *EF-1α*, *18s rRNA*, *MDH*, *THFS*, *TMp1* and *SAND* were more suitable as internal reference genes in pericarp development. The combination of *EF1-α* and *EF1-γ* was selected for grape pericarp, while *β-ACT* and *SAND* were selected as the internal reference genes in the later stage of pericarp development [65,66]. The first report to obtain stable reference genes for normalizing gene expression of abiotically stressed tissues in *E. japonica* included *GAPDH*, *EF1α* and *ACT* for floral development; *GAPDH*, *UBCE* and *ACT* for fruit setting; and *EF1α*, *GAPDH* and *eIF2B* for fruit ripening [67]. The selection of endogenous reference genes in flesh was also divided into two different endogenous reference genes in flesh and flesh development. In flesh, *EF-1α*, *CKL* and *WD40* [62] and *GAPDH* and *Mn-SOD* [63] were selected as suitable reference genes for apple and longan, respectively. In loquat, *RPL18*, *GAPDH*, *TIP41*, *EF1α*, *GAPDH* and *eIF2B* were found to be the most stable reference genes during the fruit development of loquat [67,68].

Indeed, the selection of internal reference genes during pulp development has been the subject of numerous research endeavors in various fruit species. Notably, studies conducted in banana, citrus, pear, longan, apple, lotus mist and others have presented diverse choices of reference genes (Table 3). These findings highlight the importance of carefully considering the specific species and experimental requirements when screening internal reference genes. For instance, *18s RNA* and *RPS2* are suitable internal reference genes in banana [69], while in citrus, *TUA3* and *GAPDH* were identified as appropriate choices [70]. *TUB2* is a reliable reference gene at cell division stage of pear fruits [71]. In apple, *MDH*, *SAND*, *THFS*, *TMp1*, and *WD40* are suitable internal reference genes [60]. Tong et al. [43] and You et al. [72] found that *TEF2*, *UBQ10*, *RPII*, and *ACT* were more suitable as internal reference genes in fruit development. Such variations in the selection of reference genes demonstrate the necessity of tailoring our approach to each fruit species and experimental conditions to ensure the accuracy and reliability of tested results. As conscientious researchers, we should embrace the diversity in internal reference genes, recognizing that different fruit species may require distinct reference genes for accurate gene expression analysis. By adopting this approach, we can strengthen the robustness and integrity of our research outcomes, contributing to a comprehensive understanding of fruit pulp development and its underlying mechanisms.

Research on the selection of internal reference genes during pulp development in various fruits has yielded diverse findings. Notably, different fruit species exhibit unique gene expression dynamics, necessitating the identification of suitable internal reference genes tailored to specific experimental conditions. For instance, in banana, *CAC* and *SAMDC1* were identified as appropriate reference genes [69], while citrus fruits showed *FBOX*, *SAND*, *UPL7*, and *GAPC2* to be suitable for crystal orange and sweet orange, respectively [56]. In plum fleshes and peels, Kim et al. [73] revealed *SAND protein-related trafficking protein* (*MON*), *EF-1α* and *initiation factor 5A* (*IF5A*) as the best reference genes.

The selection of reference genes in kiwifruit also varied among different varieties. Zhao et al. [74] found *ACTB* to be the most suitable for ‘Xuxiang’ kiwifruit, whereas the *GAPDH* and *UBQ* combination was suitable for various other kiwifruit varieties [46]. Similarly, ‘Jinkui’ kiwifruit young fruits demonstrated *ACT* as the most appropriate reference gene [50]. In grapes, Upadhyay et al. [75] identified *PP2A*, *SAND*, and *Sutra* as suitable internal reference genes. For fig fruits, *18s rRNA* was selected as the reference gene [47], while starfruit showed *α-Tub* and *β-Tub* as the preferred internal reference genes [53]. During fruit development, the choice of reference genes varied among different species and varieties. Comprehensive analyses have revealed that *ACT*, *UBQ*, *GAPDH*, *18s rRNA*, and *Tub* serve as suitable internal reference genes for various fruit trees during their developmental stages. In Asian pear cultivars, Chen et al. used genome-wide identification and found superior reference genes *BPS1* and *ICDH1* for transcript normalization during analyses of flesh development [76]. In vegetative tissues and organs of cherry, the best normalization was achieved with a combination of *CYP2*, *α-Tub*, *SAND-2*, and *RPL13*, as determined by geNorm software, or *RPL13*, as determined by NormFinder [77]. In pear, *SOX2* and *PP2A* were found to serve as suitable internal reference genes by Wang et al. [78]. However, it is important to emphasize that the applicability of internal reference genes is not universal across species. Therefore, diligent verification and screening of the most appropriate internal reference genes are essential for each specific fruit species.

In summary, the quest for ideal internal reference genes in fruit research is crucial for ensuring the accuracy and reliability of gene expression analyses, ultimately contributing to a comprehensive understanding of fruit pulp development and its underlying mechanisms.

Flowers mark the crucial transition of plants from vegetative to reproductive organs, and the selection of suitable internal reference genes for flower organs has been thoroughly analyzed and screened across various plant species. For citrus flowers, *TUA3*, *GAPDH*, *FBOX*, *GAPC2*, *SAND*, *UPL7*, *18s rRNA* and *RPII* were identified as potential internal reference genes [40,56,70]. In peach flowers, *TEF2*, *UBQ10*, and *RPII* were appropriate internal reference genes for subsequent gene expression analyses [79]. Similarly, kiwifruit flowers exhibited different reference genes, such as *Tub* and *ACTB* combination and *GAPDH* and *UBQ* combination [45,46]. In apple flowers, Pâmela Perini et al. [60] and Zhou et al. [59] respectively identified *MDH*, *SAND*, *THFS*, *TMp1*, *WD40*, and *UBQ* as suitable internal reference genes. For grape flowers, Upadhyay et al. [75] found *PP2A*, *SAND*, and *Sutra* to be appropriate reference genes. In citrus petals, *18s rRNA* and *RPII* were selected as the most suitable reference genes [40], while *CitUBQ14* was more stably expressed in the flower tissues of citrus at different developmental stages [80]. In pear, *WDP* and *ACT* were chosen as reference genes for pollen, style, and receptacle, respectively [57,71]. In starfruit flower buds, Li et al. [53] determined that *α-Tub* and *β-Tub* were suitable reference genes. In cherry flower bud development and dormancy release, different reference genes were selected, including *EF-1α2* and *RSP3*, *SAND-2* and *CYP2*, and *α-Tub*, *ACTB*, and *UBCE* [81,82]. For inflorescence studies, the selected reference genes were *UBQ*, *α-Tub* and *GAPDH* for jackfruit, *α-Tub1* and *β-Tub2* for durian honey, and *ACT(1*) for pitaya [35,44]. Jin et al. has found that the *RPS4* and *RPL23* combination during ovule development, and *CCR* and *RPS4* during stamen development, were sufficient for reliable normalization; this result will help facilitate the molecular breeding of pineapple for crop improvement [83]. For inflorescence development in starfruit, *β-Tub* and *UBC4* were identified as suitable internal reference genes [53].

Regarding seeds, research on the selection of internal reference genes is relatively limited. Niu et al. [84] identified *UBC* as the internal reference gene for apricot seeds. For apple seeds, Zhou et al. [59] suggested *UBQ* as the most suitable internal reference gene in their screening study.

Changes in gene expression were also detected during postharvest storage, so it is necessary to screen for endogenous reference genes during postharvest storage. Among them, different combinations of endogenous reference genes, *18S rRNA* + *EF-1α* and *18S rRNA + ACT* were screened at 4 °C and 22 °C in longan, respectively [63]. At the same time, the optimal internal control genes existed under different conditions during the storage of plum: CAC and UNK under room temperature, and *CAC*, *ACT*, and *CLATH* under cold treatment [85]. *EF-1α* exhibited the highest stability in soursop fruits stored at 15 ± 1 °C [86]. *UBQ-CONJ-E2* and *TUB-FCB* were the two best reference genes identified from kiwifruit fruits during postharvest storage [87]. *PpeIF-1A* was the most stable gene during different storage processes (5 °C, 15 °C, 25 °C, ambient temperature, and 35 °C) in peach fruits [88]. In papaya, Zhu et al. [89] found that *EIF* and *RPS* were the most suitable internal reference genes under different storage temperatures. In the process of studying the postharvest browning of litchi peel, the most stable gene was *HDAC9* [90]. 

As we delve into the study of edible fruits, with a particular emphasis on their quality determined by intrinsic gene expression, the significant role of internal reference genes in gene expression analysis comes to the forefront. This has spurred a dedicated effort among researchers to meticulously screen internal reference genes. The stability and elevated expression levels of these identified reference genes are paramount, enabling an effective showcase of the expression levels of other genes in a thoughtful and comprehensive manner.

In conclusion, the process of selecting internal reference genes for various plant reproductive organs is notably species-specific and contingent on the experimental context. It is imperative to exercise meticulous screening to identify the most suitable internal reference genes. This practice is essential to guarantee the precision and reliability of gene expression analyses, especially when studying diverse plant developmental stages and organs.

**Table 3 ijms-25-01142-t003:** Selection of reference genes in reproductive organs of fruit trees.

Species	Genus	Reproductive Organs	Reference Genes	References
Apple	*Malus*	Flowers, pericarp and pulp development process	*MDH*, *SAND*, *THFS*, *TMp1*, *WD40*	[60]
		Flowers, fruit development process, seeds	*UBQ*	[59]
		Pericarp	*ACT*, *GAPD*, *WD40H*	[62]
		Pericarp development process	*EF-1α*, *18s rRNA*	[59]
		Flesh	*EF-1α*, *CKL*, *WD40*	[64]
Apricot	*Prunus*	Fruit postharvest	*CAC* and *UNK* or *CAC*, *ACT* and *CLATH*	[85]
		Seeds	*UBC*	[84]
Cherry		Flower bud development process	*EF-1α2*, *RSP3*	[81]
		Flower bud dormancy removal process	*ACTB*, *UBCE*	[82]
Peach		Fruit development process	*ACT*	[72]
		Flowers, fruit development process	*TEF2*, *UBQ10*, *RPII*	[79]
		Fruit postharvest	*PpeIF-1A*	[88]
Plum		Fruit development process	*IPGD*, *HAM1*, *SNX1*	[73]
Soursop	*Annona*	Fruit postharvest	*EF-1α*	[86]
Banana	*Musa*	Fruit	*CAC*, *SAMDC1*	[69]
		Pulp development process	*18s rRNA*, *RPS2*	
Pomelo		Fruit development process	*β-Tub*	[42]
Citrus	*Citrus*	Flowers, flesh development process	*TUA3*, *GAPDH*	[70]
		Flower organs (petals), pericarp	*18s rRNA*, *RPII*	[40]
		Flowers, fruit	*FBOX*, *GAPC2*, *SAND*, *UPL7*	[56]
Durian honey	*Artocarpus*	Inflorescence	*α-Tub1*, *β-Tub2*	[53]
Fig	*Ficus*	Fruit	*18s rRNA*	[47]
Grape	*Vitis*	Flowers, fruit	*PP2A*, *SAND*, *Sutra*	[75]
		Fruit	*EF1-γ* and *PPR2* combination	[65]
		Late development of the pericarp	*β-ACT*, *SAND*	[66]
		Pericarp	*EF1-α* and *EF1-γ* combination	[65]
Jackfruit	*Artocarpus*	Fruit development process	*UBQ*, *GAPDH*, *18S rRNA*	[35]
		Inflorescence	*UBQ*, *GAPDH*, *α-Tub*	
Jujube	*Ziziphus*	Flowers, fruit development process	*ZjH3*	[46]
Kiwifruit	*Actinidia*	Flowers	*Tub* and *ACTB* combination	[45]
		Flowers, fruit	*GAPDH* and *UBQ* combination	[46]
		Fruit	*ACTB*	[74]
		Fruit (young fruit)	*ACT*	[50]
		Fruit postharvest	*UBQ-CONJ-E2*, *TUB-FCB*	[87]
Longan	*Dimocarpus*	Pericarp	*GAPDH*, *Fe-SOD*, *Cu/Zn-SOD*	[63]
		Pericarp development process	*EF-1α*, *Mn-SOD*	
		Pulp	*GAPDH*, *Mn-SOD*	
		Fruit postharvest	*18S rRNA* + *EF-1a* or *18S rRNA + ACT*	
Lychee	*Litchi*	Fruit development process	*β-ACT*	[58]
		Fruit postharvest	*HDAC9*	[90]
Loquat	*Eriobotrya*	Fruit development process	*EF1α*, *GAPDH*, *eIF2B*	[67]
		Fruit setting	*GAPDH*, *UBCE*, *ACT*	
		Flowers development	*GAPDH*, *EF1α*, *ACT*	
Pear	*Pyrus*	Floral organs (pollen, style)	*WDP*	[57]
		Flower organs (receptacle)	*ACT*	[71]
		Pericarp development process	*Tub2*	
		Fruit development process	*SOX2*, *PP2A*	[78]
		Pulp development process	*BPS1* and *ICDH1*	[76]
Pineapple	*Ananas*	Ovule development	*RPS4* and *RPL23* combination	[83]
		Stamen development	*CCR*, *RPS4*	
Starfruit	*Averrhoa*	Flower buds, fruit	*α-Tub*, *β-Tub*	[53]
		Inflorescence development process	*β-Tub*, *UBC4*	

### 4.3. Selection of Internal Reference Genes under Stresses

The resistance of fruit trees has always been a focus of research, and the research on resistance genes is also a priority among priorities, and the premise is to ensure the accuracy and reliability of internal reference genes. To date, reference genes have been extensively screened under stress conditions (Table 4).

For temperature stress, reference genes have been screened in banana, citrus, pear, longan, lotus mist, persimmon, cherry, and other fruit species. Chen et al. [57] selected *Tub* and *WDP* as the internal reference genes in ‘Dangshanyu’ pear at low temperature, and *UBQ* was selected as the internal reference gene under high temperature stress. Similarly, in cherry, the internal reference gene, i.e., *GAPDH* [81] was selected differently under low and high temperature stresses. In banana, *ACT1* and *EIF5A-2* were the most suitable reference genes under low and high temperature stress [69]. In longan [63], the suitable reference genes under low temperature stress are *18s rRNA*. Wei et al. [90] found that *CYP20-1* and *UBQ* were suitable internal reference genes under low temperature stress in the study of lotus mist. In *E. japonica*, *ACT*, *EF1α* and *UBCE* for leaves under heat stress and *eIF2B*, *UBCE* and *EF1α* for leaves under freezing stress are suitable combinations of reference genes [67]. According to the study of Wang et al. [91], the reference genes for persimmon under low and high temperature stresses are *UBC*, *RPII* and *Tua*. The most stable genes are *ACT* and *UBQ10* in peaches under chilling stress, thus providing guidelines for more accurate RT-qPCR results [92].

Chen et al. [57] and Zhang et al. [93] found that *GAPDH*, *β-Tub* and *UBQ* were screened as reference genes in pear under salt stress conditions. However, it was found that *TIP41* was the most stable reference gene in pear after treatments of various hormones (ABA, 6-BA and NAA) [94]. Zhu et al. [81] also found different suitable reference genes for cherries under salt stress, including *ACTB*, and *UBCE*. The internal reference genes for persimmon were *α-Tub*, *PP2A* under hormone treatment (GA/ABA/SA) and cold, heat and salt stresses [91].

In the context of disease infection, the choice of internal reference genes displays significant variability across different species and diseases. Comprehensive investigations have been carried out on various fruits, encompassing banana, mulberry, grape, citrus, mango, and peach among others. As an illustration, in the case of banana infected with banana anthracnose, studies have pinpointed *ACT1* and *EIF5A-2* as suitable reference genes for accurate gene expression analysis. This exemplifies the need for species-specific and disease-specific reference gene selection to ensure the reliability of such analyses [69]. Deng et al. [95] selected 14 candidate reference genes from a blueberry transcriptome database and used three algorithms to evaluate the expression stability of these genes under five abiotic stress conditions; then, *EF1α*, *EIF* and *TBP* were observed to be the most stable and were chosen as reference genes for qRT-PCR. *GST1* and *Tub* were identified as suitable reference genes for mulberry infected with *Sclerotinia* [96]. For grape, *EF-1a*, *SAND*, *SMD3*, *UBC*, *VAG*, and *PEP* were used as reference genes in the process of scab, mucor, and downy mildew [97,98]. In the study by Ye et al., *HISTH4*, *ACTIN2*, *DBP*, and *GAPDH*, respectively were found internal reference genes when strawberry (*Fragaria × ananassa*) seedlings were subjected to different stress conditions including heat, cold, drought, and salt [99]. Galimba et al. used RefFinder to evaluate the expression stability of *IPGD*, *HAM1* and *SNX1* as suitable internal reference genes [100]. In citrus, diseases like canker disease and fading disease require different internal reference genes, including *FBOX*, *GAPC2*, *SAND*, *UPL7*, *18s rRNA*, *ATCB*, *RPII*, and *18s rRNA* [40,56]. *GAPDH* and *gyrβ* were identified as suitable internal reference genes when the mango was infected with keratosis [101]. Xu et al. [102] screened *CYP2* and *Tua5* as appropriate reference genes when peach was infected with the tobacco crackling virus. Under the treatment of various hormones, the selection of internal reference genes in fruit trees also varies. In persimmons treated with GA, ABA, and SA hormones, *α-Tub* and *PP2A* were identified as internal reference genes for subsequent gene analyses [91]. *ACT1* and *UBQ* were found to be stably expressed and selected as the most suitable reference gene combination when kiwifruit was treated with MT/PP1/PP2/HBR1/HBR2/MT + PP/MT + HBR [46]. For longan treated with NAA and ETH, *GAPDH* and *EF-1α* were identified as appropriate internal reference genes [44]. *β-ACT* and *GAPDH* were selected as reference genes when litchi was treated with ABA, CPPU, and NAA [58,90]. *UBQ2* and *RAN* were screened as suitable internal reference genes when banana was treated with SA and MeJA [69].

Moreover, under the condition of wound stress in grape, *UBC*, *VAG*, and *PEP* were selected as internal reference genes, while under shoot pinching, SAND and *VAG* were chosen as internal reference genes [97,103]. Under shading conditions in both pear and litchi, *EF-1α* was selected as the internal reference gene [58,104]. Additionally, in pear under shading treatment, *His3* was identified as another suitable internal reference gene [104]. The diversity in the selection of internal reference genes under different conditions underscores the importance of careful and species-specific screening to ensure the accuracy and reliability of gene expression analysis.

A diverse array of genes have been identified through the systematic examination of internal reference genes in various fruit tree species experiencing diverse stress conditions. These encompass well-known candidates like *GAPDH*, *EF-1α*, *ACT*, *UBQ*, and others, along with newly discovered entities such as *DBP*, *HISTH4*, *gyrβ*, etc. Notably, these findings underscore pronounced differences in the deployment of internal reference genes under different stress conditions and across various species. Thus, the ongoing journey of internal reference gene screening indicates the necessity for continued exploration and understanding in this dynamic field.

**Table 4 ijms-25-01142-t004:** Selection of reference genes in fruit trees under stress treatments.

Species	Genus	Stress Treatments	Reference Genes	References
Banana	*Musa*	Heat and cold stresses, infection with germs (banana anthracnose)	*ACT1*, *EIF5A-2*	[69]
		Hormone treatment (SA/MeJA)	*UBQ2*, *RAN*	
Blueberry	*Vaccinium*	Salt treatment, alkaline treatment, saline–alkaline treatment, drought treatment and AlCl_3_ treatment	*EF1α*, *EIF*, *TBP*	[95]
Citrus	*Citrus*	Infection with germs (citrus bacterial canker)	*ATCB*, *18s rRNA*, *RPII*	[40]
		Infection with pathogens (*Alternaria alternata*, *Phytophthora parasitica*, *Xylella fastidiosa* and *Candidatus Liberibacter asiaticus*)	*FBOX*, *GAPC2*, *SAND*, *UPL7*	[56]
Grape	*Vitis*	Shoot pinching	*SAND*, *VAG*	[103]
Kiwifruit	*Actinidia*	Hormone treatment (MT/PP1/PP2/HBR1/HBR2/MT + PP/MT + HBR)	*ACT1*, *UBQ*	[46]
Longan	*Dimocarpus*	Cold stress	*18s rRNA*, *EF-1α*, *Fe-SOD*	[63]
		Hormone treatment (NAA/ETH)	*GAPDH*, *EF-1α*	
Lychee	*Litchi*	Hormone treatment (NAA)	*GAPDH*	[58]
		Hormone treatment (ABA/CPPU)	*β-ACT*	[90]
		Shading treatment	*EF-1α*	[58]
Loquat	*Eriobotrya*	Heat stress	*ACT*, *EF1α* and *UBCE*	[67]
		Freezing stress	*eIF2B*, *UBCE* and *EF1α*	
		Salt stress	*EF1α*, *TUA* and *UBCE*	
Mango	*Mangifera*	Infection with germs (keratosis)	*GAPDH*, *gyrβ*	[101]
Mulberry	*Morus*	Infection with a virus (sclerotinia)	*GST1*, *Tub*	[96]
Cherry	*Prunus*	Cold and salt stresses	*GAPDH*	[81]
		Hormone treatment (ABA)	*ACTB*, *UBCE*	
Peach		Chilling stress	*ACT* and *UBQ10*	[92]
		Infectious bacteria (tobacco crackling virus)	*CYP2*, *Tua5*	[102]
Pear	*Pyrus*	Cold stress	*Tub*, *WDP*	[57]
		Heat and salt stresses	*UBQ*	
		Salt stress	*GAPDH*, *β-Tub*	[93]
		Shading treatment	*EF-1α*, *His*	[104]
		hormone treatments (ABA, 6-BA and NAA)	*TIP41*	[94]
Persimmon	*Diospyros*	Cold, heat and salt stresses	*UBC*, *RPII*, *Tua*	[91]
		Hormone treatment (GA/ABA/SA)	*α-Tub*, *PP2A*	
Pitaya	*Hylocereus*	Cold stress	*ACT(1)*	[44]
Strawberry	*Fragaria*	Heat stress	*HISTH4*	[99]
		Cold stress	*ACTIN2*	
		Drought stress	*DBP*	
		Salt stress	*GAPDH*	

### 4.4. Application of Internal Reference Genes in Fruit Trees

At present, many studies on fruit trees are based on morphological characterization and physiology, but when it comes to its underlying mechanism, we need to know whether there are differences and connections in gene expression, and in the determination of gene expression, internal reference genes occupy a pivotal position. Any study of gene expression requires an internal control to be able to normalize the expression level of the gene. At present, many internal reference genes have been reported and applied successively, but there are still many conditions and species applicable to internal reference genes that have not been discovered, which is also the next research direction and provides a good basis for follow-up experiments.

Different internal reference genes can be applied under different experimental conditions. *ACT*, *β-actin*, *EF-1α* and *rRNA* are the most commonly used internal reference genes in the study of gene expression in fruit trees. For example, *ACT* can be used as an internal reference gene to verify gene expression levels in apples during storage [105,106], and it is also used as the internal reference gene in the study of nematode resistance in *Prunus* spp. [107]. *ACT2* is used to standardize the detection of the transcription level of qPCR products during the storage of pears [108]. *β-actin* was used as an internal reference gene to analyze the cuticular waxes and related gene expression between ‘Newhall’ and ‘Ganqi 3’ navel oranges during long-term cold storage [109]. *β-actin* is also used as an internal reference gene to characterize organic acid metabolism-related genes during the fruit development of *Actinidia eriantha* [110]. In the study of gray mold in strawberry, *EF-1α* was selected as the internal reference gene for the relative expression analyses of the target genes [111]. *EF-1α* was also used as an internal reference gene to detect rootstock effects on anthocyanin accumulation and associated biosynthetic gene expression during fruit development and ripening of blood oranges [112]. The *25-s rRNA* was used as an internal reference gene for gene expression activities of guava during low-temperature storage after 1-MCP treatment [113]. In ripening stages of ‘Siam Red Ruby’ fruit (*Citrus grandis*), *18S rRNA* was used as the internal reference gene to investigate the changes in the accumulation of carotenoid and carotenogenic gene expressions [114].

## 5. Conclusions and Future Perspectives

The selection of internal reference genes in fruit trees represents a pivotal stage in gene expression analysis. Nonetheless, the quest for the perfect reference gene, one that maintains stable expression levels across all tissues, developmental stages, and physiological conditions, remains an ongoing challenge and has yet to yield a definitive solution. The stability and reliability of endogenous reference genes are relative and can vary significantly across different experimental conditions and species. Researchers must carefully choose suitable internal reference genes based on the specific sample types and experimental conditions.

The expression stability of candidate reference genes can be assessed using qRT-PCR, and selection can be made from various sources, such as traditional housekeeping genes, newly discovered stable genes in other plants, and the application of transcriptome sequencing technology [115]. Additionally, gene expression chips and EST databases offer promising resources for identifying new candidate genes. It is essential to verify the stability of selected reference genes through multiple evaluation software tools to ensure the accuracy of tested results [79]. This systematic approach helps to account for the differences in stability among various reference genes in different fruit varieties, tissues, and experimental conditions. However, there are many researchers who still use commonly used genes as internal reference genes, so we still need to find more accurate and effective methods on the road of mining new internal reference genes.

The measurement of gene expression in fruit trees under various conditions is of great significance. According to the existing reports, there are often great differences in the optimal reference genes of different species under different conditions. The reliability of reference genes directly impacts the accuracy of gene expression analysis. Although the systematic verification of reference gene stability is still in its early stages, it provides a vital reference standard for qRT-PCR research and contributes to understanding the internal mechanisms of gene expression.

In this comprehensive review, we delve into the criteria and assessment methodologies guiding the selection of candidate reference genes, coupled with recent strides in reference gene research within the realm of fruit trees. Our goal is to furnish researchers in this field with invaluable insights into the nuanced process of identifying candidate reference genes and leveraging stable references, thereby ensuring the precision and reliability of gene expression analyses. As molecular biotechnology advances and evolves, an increasing number of stable and well-suited candidates for quantitative experiments are likely to be identified. Concurrently, the progress in information technology and data integration open the possibility of establishing an information-sharing platform. This platform can serve the purpose of consolidating and summarizing the most appropriate reference genes for specific tissues, species, and conditions in fruit trees, facilitating real-time PCR applications.

## Figures and Tables

**Figure 1 ijms-25-01142-f001:**
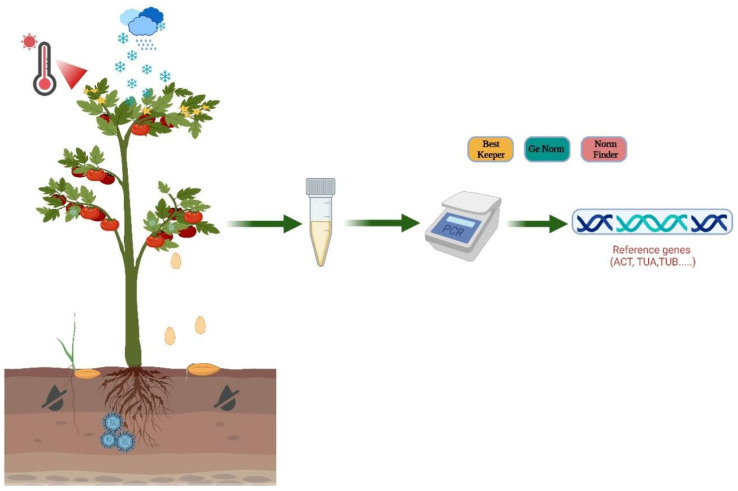
qRT-PCR illustration model.

**Table 1 ijms-25-01142-t001:** Commonly used reference genes and their functions in plants.

Gene Symbols	Full Names	Functions
*ACT*	*Actin*	An important skeleton protein of the cell
*EF-1α*	*Elongation factor* *-1α*	Elongation of transcription
*GAPDH*	*Glyceraldehyde-3-phosphate-dehydrogenase*	Key enzymes in the carbon fixation cycle of glycolysis, gluconeogenesis, and photosynthesis
*His*	*Histone*	Formation of higher chromosome structures
*β-ACT*	*Beta actin*	Maintenance of cellular structure, intracellular movement, and cell division
*18s rRNA*	*18s ribosomal RNA*	Cytoplasmic ribosome small subunit, translation
*α-Tub*	*Alpha tubulin*	Cytoskeletal structural proteins
*β-Tub*	*Beta tubulin*	Cells grow and participate in light-stimulating responses
*UBC*	*Ubiquitin conjugating enzyme*	Label proteins that need to be broken down, causing them to hydrolyze
*UBQ*	*Ubiquitin*	Protein modification, binding, and degradation

## Data Availability

Not applicable.

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
