# Peer review of "Advancements in Reference Gene Selection for Fruit Trees: A Comprehensive Review"

_ijms, 2024, doi:10.3390/ijms25021142_

Round 1

Reviewer 1 Report

Comments and Suggestions for Authors

Thank you for sending me this interesting manuscript by Shujun Peng, Irfan Ali Sabir, et al.  Overall, this study intends to provide a definitive reference the upcoming research that will be conducted on fruit trees. The authors summarized the research progress in recent years regarding the selection and stability analysis of the candidate reference genes of different fruit trees.

Overall, this manuscript provides a good description and detailed information in reference gene selection and the state of art in the field.

There are minor issues that are listed in order, as follows:

1) provide a higher resolution Figure 1.

2) the section conclusion and perspectives need to be improved, please add more information to this section.

3) page 2, line 71: the authors mention the types of RNAs (tRNA and rRNA) but do not specify examples of those RNAs, it would be interesting to include more details on this section.

4) although one of the goals of this manuscript is to provide diverse types of potential reference genes for fruit three, most of the reference genes described in this review are common ones, like variant of tubulin or actin.

5) The authors give detail information in very specific fruit trees, please justify the reason to choose these examples.

Author Response

Thank you for sending me this interesting manuscript by Shujun Peng, Irfan Ali Sabir, et al.  Overall, this study intends to provide a definitive reference the upcoming research that will be conducted on fruit trees. The authors summarized the research progress in recent years regarding the selection and stability analysis of the candidate reference genes of different fruit trees.

Overall, this manuscript provides a good description and detailed information in reference gene selection and the state of art in the field.

There are minor issues that are listed in order, as follows:

1) Provide a higher resolution Figure 1.

Response: Thanks for your advice. We have improved the Figure 1 quality. Please check the updated version of manuscript.

2) The section conclusion and perspectives need to be improved, please add more information to this section.

Response: Thank you so much for your keen and critical review for manuscript and for your advice. The section of conclusion and perspectives has been improved and more information has now been added to this section. Please check the updated version of manuscript.

3) Page 2, line 71: the authors mention the types of RNAs (tRNA and rRNA) but do not specify examples of those RNAs, it would be interesting to include more details on this section.

Response: Thanks for your advice. We have added specify RNAs (tRNA and rRNA) information in updated version of manuscript.

4) Although one of the goals of this manuscript is to provide diverse types of potential reference genes for fruit three, most of the reference genes described in this review are common ones, like variant of tubulin or actin.

Response: Thanks for your advice. At present, the housekeeping gene is still the main reference gene selected for fruit trees, because its expression level will still be more suitable as the internal reference gene than other genes in research experiments. However, at the same time, many new reference genes have been screened under specific conditions.

5) The authors give detail information in very specific fruit trees; please justify the reason to choose these examples.

Response: Thanks for your advice. The internal reference genes relevant to various species within the same family and genus may not be identical. Consequently, when investigating diverse species, researchers undertake the verification and screening of distinct internal reference genes. This underscores the importance of synthesizing a wealth of studies to identify and compile the species that have been examined.

Reviewer 2 Report

Comments and Suggestions for Authors

The manuscript (review) is edited on a current topic and in many cases contains up-to-date information on the given area. Nevertheless, I would recommend some adjustments to the authors. The actual txt about reference genes is divided into three parts, and at the end of each of them it would be good to include a short paragraph about the given regions in fruit species and other tree species for a more comprehensive view of the given issue. E.g. a very interesting study by Su et al. (2023) /DOI: 10.17221/72/2022-CJGPB/ describes the comprehensive use of reference genes in Paulownia fortunei, an important fast-growing woody species for biomass production. In section 4.2. are devoted to the generative parts of fruits, but there is a complete lack of a section devoted to studies during fruit storage, which is also a very important area and I recommend section 4.2. expand on this. E.g. the use of actin in the storage of apples is described by Zdarska and Cmejla (2023) /DOI: 10.17221/102/2022-CJGPB/ and similar studies were also described for other fruit species. Based on the above comments and notes, I recommend the manuscript for publication after major revision and second review.

Author Response

The manuscript (review) is edited on a current topic and in many cases contains up-to-date information on the given area. Nevertheless, I would recommend some adjustments to the authors. The actual txt about reference genes is divided into three parts, and at the end of each of them it would be good to include a short paragraph about the given regions in fruit species and other tree species for a more comprehensive view of the given issue. E.g. a very interesting study by Su et al. (2023) /DOI: 10.17221/72/2022-CJGPB/ describes the comprehensive use of reference genes in Paulownia fortunei, an important fast-growing woody species for biomass production.

In section 4.2. are devoted to the generative parts of fruits, but there is a complete lack of a section devoted to studies during fruit storage, which is also a very important area and I recommend section 4.2. expand on this. E.g. the use of actin in the storage of apples is described by Zdarska and Cmejla (2023) /DOI: 10.17221/102/2022-CJGPB/ and similar studies were also described for other fruit species.

Based on the above comments and notes, I recommend the manuscript for publication after major revision and second review.

Response: Thank you for your guidance. Initially, we have categorized the three segments and concentrated on elucidating the connection between the relevant reference genes and their corresponding organs. Subsequently, incorporating your valuable suggestions, we have expanded our scope to include an analysis of the internal reference genes identified during the postharvest storage of fruit trees.

Round 2

Reviewer 2 Report

Comments and Suggestions for Authors

The authors did not deal with the comments in a fundamental way. Although the text has been modified, it now includes a section dedicated to post-harvest ripening, but I consider it appropriate to supplement the tables that demonstrate the different uses of reference genes. Here, these tables would deserve more references. Currently, only those that are part of the text are listed. It is the addition of references in the tables for the already mentioned species that would significantly increase the scientific value of the evaluated review. E.g. in Table 3, the gene for actin in post-harvest ripening is missing (Malus - Zdarska and Cmejla /2023 - DOI: 10.17221/102/2022-CJGPB/ or in Table 4, where stresses are presented, there is no information on biotic stress (resistance to nematodes) in important representative of Prunus cerasifara (rootstock) Liu et al. /2023 - DOI: 10.17221/111/2022-CJGPB/, where the gene for actin was also used. I consider it essential to supplement the tables with additional references, because when looking at the Web of Science database there are enough articles for 2023 alone. At the same time, I consider it appropriate that the authors also mark any newly inserted references in the References section, as this would facilitate the search and identification of modifications (which is very difficult in version 2). I recommend the manuscript for publication after major revision and second review.

Author Response

The authors did not deal with the comments in a fundamental way. Although the text has been modified, it now includes a section dedicated to post-harvest ripening, but I consider it appropriate to supplement the tables that demonstrate the different uses of reference genes. Here, these tables would deserve more references. Currently, only those that are part of the text are listed. It is the addition of references in the tables for the already mentioned species that would significantly increase the scientific value of the evaluated review. E.g. in Table 3, the gene for actin in post-harvest ripening is missing (Malus - Zdarska and Cmejla /2023 - DOI: 10.17221/102/2022-CJGPB/ or in Table 4, where stresses are presented, there is no information on biotic stress (resistance to nematodes) in important representative of Prunus cerasifara (rootstock) Liu et al. /2023 - DOI: 10.17221/111/2022-CJGPB/, where the gene for actin was also used. I consider it essential to supplement the tables with additional references, because when looking at the Web of Science database there are enough articles for 2023 alone. At the same time, I consider it appropriate that the authors also mark any newly inserted references in the References section, as this would facilitate the search and identification of modifications (which is very difficult in version 2). I recommend the manuscript for publication after major revision and second review.

Our response: Thanks for your advice. In the revised manuscript, seventeen new references were added in Reference section (marked with red color) and a paragraph was provided about application of internal reference genes in fruit trees.

Round 3

Reviewer 2 Report

Comments and Suggestions for Authors

The authors have accepted all my comments and the manuscript can be accepted for publication.